# Thrombopoietin from hepatocytes promotes hematopoietic stem cell regeneration after myeloablation

**Longfei Gao[1,2,3†], Matthew Decker[1,2,3†], Haidee Chen[1,2,3], Lei Ding[1,2,3]***

[1]Columbia Stem Cell Initiative, Columbia University Medical Center, New York, United States; [2]Department of Rehabilitation and Regenerative Medicine, Columbia University Medical Center, New York, United States; [3]Department of Microbiology and Immunology, Columbia University Medical Center, New York, United States, New York, United States

**Abstract** The bone marrow niche plays critical roles in hematopoietic recovery and hematopoietic stem cell (HSC) regeneration after myeloablative stress. However, it is not clear whether systemic factors beyond the local niche are required for these essential processes in vivo. Thrombopoietin (THPO) is a key cytokine promoting hematopoietic rebound after myeloablation and its transcripts are expressed by multiple cellular sources. The upregulation of bone marrow-derived THPO has been proposed to be crucial for hematopoietic recovery and HSC regeneration after stress. Nonetheless, the cellular source of THPO in myeloablative stress has never been investigated genetically. We assessed the functional sources of THPO following two common myeloablative perturbations: 5-fluorouracil (5-FU) administration and irradiation. Using a *Thpo* translational reporter, we found that the liver but not the bone marrow is the major source of THPO protein after myeloablation. Mice with conditional *Thpo* deletion from osteoblasts and/or bone marrow stromal cells showed normal recovery of HSCs and hematopoiesis after myeloablation. In contrast, mice with conditional *Thpo* deletion from hepatocytes showed significant defects in HSC regeneration and hematopoietic rebound after myeloablation. Thus, systemic THPO from the liver is necessary for HSC regeneration and hematopoietic recovery in myeloablative stress conditions.

**\*For correspondence:**
ld2567@cumc.columbia.edu

[†]These authors contributed equally to this work

**Competing interest:** The authors declare that no competing interests exist.

## Introduction

Hematopoietic stem cells (HSCs) generate all blood and immune cells, and play critical roles in the regeneration of the blood system after stress. They reside in the bone marrow niche where Leptin Receptor[+] (LepR[+]) perivascular stromal cells and endothelial cells are key components. These stromal cells are the major sources of factors that promote HSC maintenance (*Ding and Morrison, 2013*; *Ding et al., 2012*; *Lee et al., 2017*). Following myeloablative stress, the bone marrow niche supports HSC regeneration, which is essential for the recovery of hematopoiesis (*Zhou et al., 2015*; *Hooper et al., 2009*; *Kopp et al., 2005*; *Li et al., 2008*). Several signaling pathways, including Notch, EGF/EGFR, pleiotrophin, Dkk1, angiogenin, and SCF/c-KIT, are essential for the regeneration of HSCs and hematopoiesis (*Butler et al., 2010*; *Doan et al., 2013*; *Goncalves et al., 2016*; *Himburg et al., 2017*; *Himburg et al., 2010*; *Kimura et al., 2011*). Many of the secreted cytokines activating these pathways arise from the local bone marrow niche and regulate HSC regeneration and hematopoietic recovery in a paracrine fashion (*Guo et al., 2017*; *Himburg et al., 2017*; *Poulos et al., 2013*; *Zhou et al., 2017*). However, it is not clear whether systemic signals that originate outside the bone marrow also promote HSC regeneration and hematopoietic recovery after stress.

Thrombopoietin (THPO) is critical for bone marrow HSC maintenance and hematopoietic recovery (**Decker et al., 2018**; **Mouthon et al., 1999**; **Qian et al., 2007**; **Yoshihara et al., 2007**). It was first identified as the ligand to the MPL receptor and a driver of megakaryopoiesis and platelet production (**de Sauvage et al., 1994**; **Kaushansky et al., 1994**; **Lok et al., 1994**). Later results show that the THPO/MPL signaling is also required for the maintenance of HSCs (**Kimura et al., 1998**). Circulating levels of THPO are inversely related to platelet counts (**de Graaf et al., 2010**; **Kaushansky, 2005**), consistent with a 'receptor sink' model where HSCs are stimulated to proliferate and self-renew by free systemic THPO ligand, which is gradually soaked up by the expanding population of MPL-expressing progenies until equilibrium is reached. In line with this model, deletion of the MPL receptor from megakaryocyte-lineage cells does not lead to a loss of megakaryocytes or platelets, instead, leads to an expansion of HSCs with accompanying megakaryocytosis and thrombocytosis (**Ng et al., 2014**). By conditionally deleting *Thpo* from the bone marrow or liver, we have recently showed that steady-state HSC maintenance depends on hepatocyte-derived THPO (**Decker et al., 2018**), further highlighting the importance of systemic THPO on HSCs.

The THPO/MPL signaling also plays a crucial role in hematopoietic stress response, particularly after myeloablation. The characteristic hematopoietic progenitor rebound at around 10 days following administration of the antimetabolite drug 5-fluorouracil (5-FU) is dependent on MPL (**Li and Slayton, 2013**). After irradiation, the THPO/MPL signaling is similarly essential for hematopoietic recovery and survival (**de Laval et al., 2014**; **de Laval et al., 2013**; **Mouthon et al., 1999**; **Wang et al., 2015**). Indeed, THPO mimetic drugs, such as romiplostim and eltrombopag, have been shown to improve recovery after ablative challenge, and have also been used clinically to support hematopoiesis in diseases such as immune thrombocytopenic purpura and aplastic anemia (**Desmond et al., 2014**; **Gill et al., 2017**; **Rodeghiero and Ruggeri, 2015**; **Yamaguchi et al., 2018**).

The regulation of THPO production has been extensively investigated, but the in vivo source of THPO for HSC regeneration and hematopoietic recovery after myeloablation is not clear. Previous studies have found that bone marrow cell populations such as stromal cells and osteoblasts may upregulate THPO in hematopoietic stress conditions, whereas the liver produces *Thpo* transcripts at a constant level (**Sungaran et al., 1997**; **Yoshihara et al., 2007**). However, other investigators have found no significant changes in bone marrow *Thpo* transcript levels after 5-FU-mediated myeloablative treatment (**Li and Slayton, 2013**). Because *Thpo* expression is under heavy translational control (**Ghilardi et al., 1998**), it is not clear what cells produce THPO protein for HSC regeneration and hematopoietic recovery after myeloablation. Furthermore, although upregulation of THPO may be a key mechanism of the bone marrow response to hematopoietic stress, the role of local THPO from bone marrow niche or systemic THPO from the liver for HSC and hematopoietic recovery has not been functionally investigated in vivo. Nonetheless, most studies proposed that local THPO derived from the bone marrow niche is critical for HSC and hematopoietic recovery after myeloablation (**Kaushansky, 2005**; **Yoshihara et al., 2007**). Here, we genetically dissected the in vivo source of THPO for HSC regeneration and hematopoietic recovery following myeloablative stress.

## Results

### Myeloablation induced by 5-FU drives THPO-dependent hematopoietic recovery and HSC expansion

5-FU is a commonly used chemotherapy agent that leads to myeloablation. To test whether THPO is required for hematopoietic recovery after 5-FU treatment, we administrated 5-FU to *Thpo* knockout (*Thpo*^gfp/gfp^) (**Decker et al., 2018**) and wild-type control mice at 150 mg/kg via a single intraperitoneal injection. Because *Thpo*^gfp/gfp^ mice have hematopoietic phenotypes compared with wild-type controls without any treatment (**Decker et al., 2018**), we also normalized hematopoietic parameters with baseline mice of the same genotype without any treatment. Ten days after the 5-FU administration, treated wild-type mice showed normal leukocyte and neutrophil counts, normal reticulocyte frequency, but significantly increased platelet counts, while treated *Thpo*^gfp/gfp^ mice had no platelet expansion (***Figure 1—figure supplement 1A-H***). When normalized to baseline levels, *Thpo*^gfp/gfp^ mice had a significant reduction of leukocyte, neutrophil, and platelet counts as well as reticulocyte frequency compared with wild-type controls (***Figure 1—figure supplement 1A-H***), suggesting that

*Thpo* is required for hematopoietic recovery. Normalized post-treatment bone marrow and spleen cellularity did not differ significantly in *Thpo^gfp/gfp^* mice compared with wild-type controls (*Figure 1—figure supplement 1I-L*). The frequencies of bone marrow and spleen HSCs (Lineage^-^ Sca-1^+^ c-kit^+^ CD150^+^CD48^-^, LSKCD150^+^CD48^-^) were increased by more than 12 fold in 5-FU-treated wild-type mice compared with baseline controls, while 5-FU-treated *Thpo^gfp/gfp^* mice showed no change from the baseline (*Figure 1—figure supplement 1M-P*). Overall, 5-FU stimulated a 3.4-fold increase of total HSC number in wild-type mice, but no significant effects were observed in *Thpo^gfp/gfp^* mice (*Figure 1—figure supplement 1Q*). Compared with wild-type controls, the HSC increase in response to 5-FU in *Thpo^gfp/gfp^* mice was reduced by 6 fold (*Figure 1—figure supplement 1R*). The effects of 5-FU on bone marrow and spleen hematopoietic progenitor cells (LSK) were also blunted in *Thpo^gfp/gfp^* mice (*Figure 1—figure supplement 1S-V*). These data confirm that hematopoietic recovery, expansion of HSCs, and hematopoietic progenitors following chemoablation mediated by 5-FU are THPO-dependent.

## Hepatic but not bone marrow THPO is required for HSC and hematopoietic recovery after 5-FU-mediated myeloablation

To identify the source of THPO, we investigated *Thpo* expression following 5-FU treatment. It has been reported that the bone marrow upregulates *Thpo* mRNA in stress conditions (*Sungaran et al., 1997*). Although bone marrow stromal cells (CD45^-^ Ter119^-^) express *Thpo* transcripts, we did not observe a significant upregulation of *Thpo* transcripts 14 days after 5-FU injection compared with baseline levels (*Figure 1—figure supplement 2A*). THPO protein production is under strong translational controls (*Ghilardi et al., 1998*). To assess the translation of THPO protein following 5-FU treatment, *Thpo^creER^; Rosa26^LSL-ZsGreen^* translational reporter mice (*Decker et al., 2018*) were injected with 5-FU and immediately started on a 10 -day course of tamoxifen treatment. Although robust expression of ZsGreen was observed in the liver (*Figure 1A*), no ZsGreen fluorescence was appreciated in the bone marrow (*Figure 1B*). Within the liver, ZsGreen were from HNF4a^+^ hepatocytes (*Figure 1C and D*). These data suggest that hepatocytes but not the bone marrow are the major source of THPO after 5-FU treatment.

Although we did not detect appreciable *Thpo* translation in the bone marrow after 5-FU treatment (*Figure 1B*), osteoblasts and LepR^+^ mesenchymal stromal cells are the major bone marrow cells expressing *Thpo* transcripts (*Decker et al., 2018*; *Guerriero et al., 1997*; *Sungaran et al., 1997*), and based on antibody staining, it has been reported that osteoblasts express THPO protein after 5-FU treatment (*Yoshihara et al., 2007*). To formally test the function of THPO from these cells, we conditionally deleted *Thpo* from osteoblasts (*Col1a1-cre; Thpo^fl/gfp^*) or bone marrow mesenchymal stromal cells (*Lepr^cre^; Thpo^fl/gfp^*). Deletion of *Thpo* from osteoblasts or LepR^+^ stromal cells had no significant effects on leukocytes, neutrophils, platelets, reticulocytes, bone marrow and spleen cellularity, HSCs, or LSKs after 5-FU treatment (*Figure 1E–M*, *Figure 1—figure supplement 2B and C*). These results show that hematopoietic recovery and HSC regeneration from 5-FU treatment do not require THPO production by bone marrow osteoblasts or mesenchymal stromal cells.

It is possible that osteoblasts and mesenchymal stromal cells serve as redundant sources of THPO. We thus also generated *Prrx1-cre; Thpo^fl/gfp^* mice. *Prrx1-cre* drives recombination in mesenchymal lineage cells in murine long bones, including both osteoblasts and mesenchymal stromal cells (*Ding and Morrison, 2013*; *Greenbaum et al., 2013*; *Logan et al., 2002*), allowing conditional deletion of *Thpo* from both osteoblasts and mesenchymal stromal cells. Compared with littermate controls, adult baseline *Prrx1-cre; Thpo^fl/gfp^* mice had normal blood cell counts, normal bone marrow cellularity, normal HSC and LSK frequencies, and showed no signs of extramedullary hematopoiesis in the spleen (*Figure 1—figure supplement 2D-N*), consistent with the notion that THPO from the bone marrow is not required for HSC maintenance or hematopoiesis (*Decker et al., 2018*). We then administrated 5-FU to these mice. Deletion of *Thpo* from both osteoblasts and mesenchymal cells in *Prrx1-cre; Thpo^fl/gfp^* mice had no effects on leukocytes, neutrophils, platelets, reticulocytes, bone marrow and spleen cellularity, HSCs, or LSKs after 5-FU chemoablation (*Figure 1E–M*, *Figure 1—figure supplement 2B and C*). These results strengthen our finding that the bone marrow is not a source of THPO for hematopoietic recovery and HSC regeneration following 5-FU treatment.

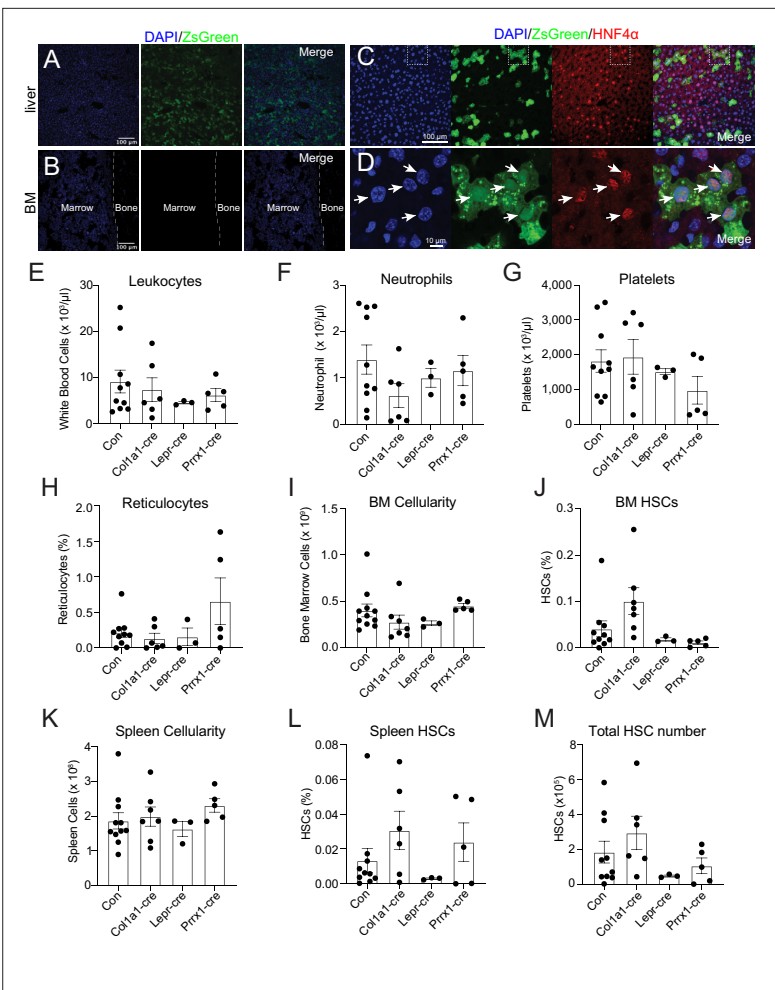

**Figure 1.** Bone marrow thrombopoietin (THPO) is not required for hematopoietic recovery and hematopoietic stem cell (HSC) expansion after 5-fluorouracil (5-FU) chemoablation. (**A**) Confocal images of liver sections from tamoxifen-treated *Thpo^creER*; *Rosa26^LSL-ZsGreen* mice 10 days after 5-FU injection. (**B**) Confocal images of femur sections from tamoxifen-treated *Thpo^creER*; *Rosa26^LSL-ZsGreen* mice 10 days after 5-FU injection. (**C and D**) Confocal images showing immunostaining of HNF4α on liver sections from tamoxifen-treated *Thpo^creER*; *Rosa26^LSL-ZsGreen* mice 10 days after 5-FU injection. (**D**) Enlarged image of the region denoted in C. Arrows point to individual HNF4α⁺ hepatocytes. (**E–H**) Blood counts of mice with *Thpo* conditionally deleted from osteoblasts (Col1a1-cre), bone marrow stromal cells (Lepr-cre), or both (Prrx1-cre), and controls after 5-FU challenge. n = 3–10 mice. (**I–L**) Cellularity and frequencies of HSCs in the bone marrow and spleens from mice with *Thpo* conditionally deleted from osteoblasts, bone marrow stromal cells, or both, and controls after 5-FU challenge. n = 3–11 mice. (**M**) Total numbers of HSCs in the bone marrow and spleens from mice with *Thpo* conditionally deleted from osteoblasts, bone marrow stromal cells, or both, and controls after 5-FU challenge. n = 3–10 mice. Con: *Thpo^gfp/+* control mice. Col1a1-cre: *Col1a1-cre; Thpo^fl/gfp* mice. Lepr-cre: *Lepr^cre; Thpo^fl/gfp* mice. Prrx1-cre: *Prrx1-cre; Thpo^fl/gfp* mice. Data represent mean ± SEM. Statistical significance was calculated with one-way ANOVA with Dunnett's test. Each dot represents one independent mouse in E–M.

The online version of this article includes the following source data and figure supplement(s) for figure 1:

**Source data 1.** Numerical values of the data plotted in panels E-M.

**Figure supplement 1.** Hematopoietic recovery and hematopoietic stem cell (HSC) expansion after chemoablation with 5-fluorouracil (5-FU) depend on thrombopoietin (THPO).

**Figure supplement 1—source data 1.** Numerical values of the data plotted in panels A-V.

**Figure supplement 2.** Hematopoietic progenitor expansion after 5-fluorouracil (5-FU) chemoablation does not depend on bone marrow thrombopoietin (THPO) and THPO from bone marrow *Prrx1*⁺ mesenchymal cells is dispensable for thrombopoiesis and hematopoietic stem cell (HSC) maintenance.

**Figure supplement 2—source data 1.** Numerical values of the data plotted in panels A-N.

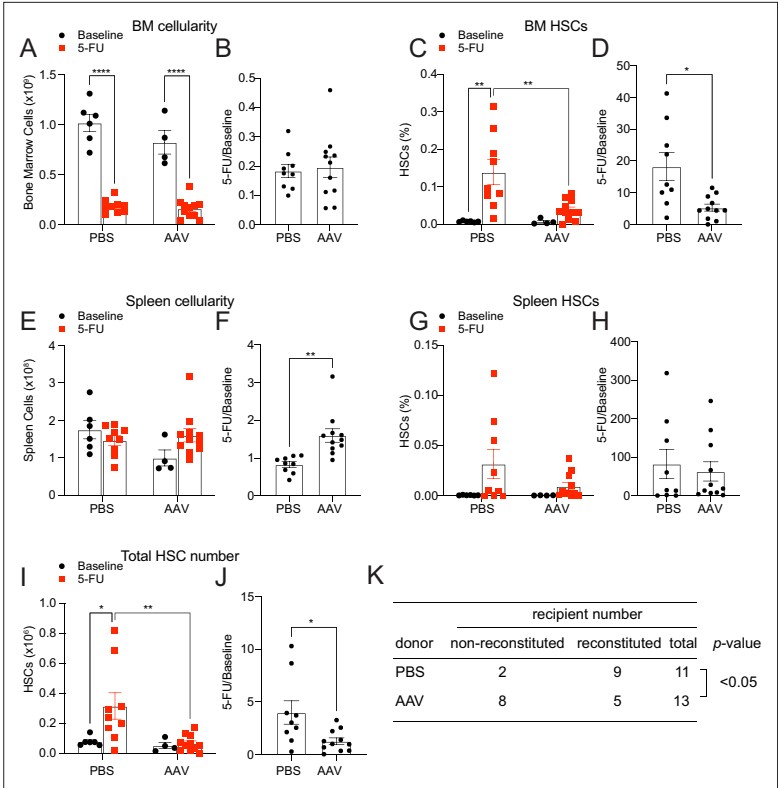

**Figure 2.** Hepatic thrombopoietin (THPO) is required for hematopoietic recovery and hematopoietic stem cell (HSC) expansion after 5-fluorouracil (5-FU) chemoablation. (**A–H**) Cellularity and frequencies of HSCs (**A, C, E, G**) in the bone marrow and spleens from mice with *Thpo* conditionally deleted from hepatocytes (AAV) and controls, with and without 5-FU challenge. Normalized fold changes (relative to no 5-FU challenge baseline) (**B, D, F, H**) are also shown. n = 4–11 mice. (**I and J**) Total numbers of HSCs (**I**) in the bone marrow and spleens from mice with *Thpo* conditionally deleted from hepatocytes and controls, with and without 5-FU challenge. Normalized fold changes (relative to no 5-FU challenge baseline) (**J**) are also shown. n = 4–11 mice. (**K**) Numbers of total recipients and long-term multilineage reconstituted recipients after transplantation of 50,000 bone marrow cells from PBS- or AAV-treated *Thpo*<sup>fl/fl</sup> and 5-FU challenged mice along with 500,000 unchallenged competitor bone marrow cells. Recipients were scored as reconstituted if they showed donor-derived myeloid, B and T cells in the peripheral blood 16 weeks after transplantation. Data were from recipients of three independent donor pairs from two independent experiments. PBS: PBS-treated *Thpo*<sup>fl/fl</sup> or wild-type control mice. AAV: AAV8-TBG-cre-treated *Thpo*<sup>fl/fl</sup> mice. Data represent mean ± SEM. *p < 0.05, **p < 0.01, ****p < 0.0001. Statistical significance was assessed with two-way ANOVA with Turkey's test (**A, C, E, G, I**), Student's t-test (**B, D, F, H, J**), or Fisher's exact test (**K**). Each dot represents one independent mouse in A–J.

The online version of this article includes the following source data and figure supplement(s) for figure 2:

**Source data 1.** Numerical values of the data plotted in panels A-J.

**Figure supplement 1.** Limited impact of acute *Thpo* deletion from hepatocytes on leukocytes and hematopoietic progenitors after 5-fluorouracil (5-FU) treatment.

**Figure supplement 1—source data 1.** Numerical values of the data plotted in panels A-M.

Hepatic THPO is required for steady-state bone marrow HSC maintenance as conditional deletion of *Thpo* from hepatocytes leads to HSC depletion (*Decker et al., 2018*). We found that hepatocytes are also the major source of THPO after 5-FU treatment (*Figure 1A–D*). To address whether hepatic THPO is required for hematopoietic recovery and HSC regeneration after 5-FU-mediated stress, we treated adult *Thpo*<sup>fl/fl</sup> mice with replication incompetent hepatotropic AAV8-TBG-cre viral vector (AAV) concurrently with 5-FU injection. A single dose of this *cre*-bearing viral vector drives efficient and specific recombination in hepatocytes (*Figure 2—figure supplement 1A* and *Decker et al., 2018*). Ten days after the treatment, AAV-treated mice showed a trend to reduction of platelet and reticulocyte expansion, with normal leukocyte and neutrophil responses to 5-FU relative to controls

(*Figure 2—figure supplement 1B-I*). Although the response of spleen HSC frequency to 5-FU was not significantly impacted by AAV treatment (*Figure 2G and H*), the responses of bone marrow HSC frequency and the total number of HSCs were significantly decreased in AAV-treated mice compared to controls (*Figure 2A–J*), indicating a compromised HSC recovery after 5-FU treatment. Of note, we observed a significant increase of the responses of spleen cellularity to 5-FU by AAV treatment (*Figure 2E and F*), indicating possible enhanced extramedullary hematopoiesis. Consistent with reduced HSC regeneration, bone marrow cells from these AAV-treated mice after 5-FU administration showed a significant reduction in reconstituting irradiated recipient mice compared with controls (*Figure 2K*). The responses of bone marrow and spleen LSK frequencies to 5-FU were largely normal in AAV-treated mice compared with controls (*Figure 2—figure supplement 1J-M*), suggesting that the effects of acutely disrupted THPO signaling are most pronounced on HSCs, rather than restricted hematopoietic progenitors. Altogether, these data suggest that hepatic but not bone marrow-derived THPO is required for hematopoietic and HSC recovery after 5-FU treatment.

## Hematopoiesis from THPO-deficient mice are sensitive to ionizing radiation

Ionizing radiation is another common myeloablative treatment. To test whether THPO is required for hematopoietic recovery in this condition, we exposed mice to a single 5.5 Gy dose of radiation. Four weeks after the irradiation, *Thpo^{gfp/gfp}* mice had significant reduction of leukocyte and neutrophil recovery with normal platelet and reticulocyte responses compared with wild-type controls (*Figure 3—figure supplement 1A-H*). *Thpo^{gfp/gfp}* mice also had significantly decreased responses to irradiation in bone marrow cellularity, LSK and HSC frequencies relative to wild-type controls (*Figure 3—figure supplement 1I-N*). The functional impact of these differences was demonstrated by a significant increase in mortality among *Thpo^{gfp/gfp}* mice following irradiation (*Figure 3—figure supplement 1O*). Compared with baseline levels, neither wild-type nor *Thpo^{gfp/gfp}* mice had notable differences in spleen cellularity, or spleen LSK frequency in response to irradiation (*Figure 3—figure supplement 1P and R*). However, *Thpo^{gfp/gfp}* mice had diminished regeneration of spleen HSC frequency and total HSC number compared with controls (*Figure 3—figure supplement 1P-W*). Thus, *Thpo* is required for hematopoietic and HSC recovery after irradiation.

## Hepatic but not bone marrow THPO is required for HSC regeneration after ionizing radiation

It is not clear what cells are the major source of THPO after irradiation. Bone marrow stromal cells (CD45^- Ter119^-) did not have significantly upregulated *Thpo* transcripts 14 days after irradiation (*Figure 1—figure supplement 2A*). To assess the translation of THPO protein following irradiation, *Thpo^{creER}*; *Rosa26^{LSL-ZsGreen}* translational reporter mice were irradiated and immediately dosed with a 10 -day course of tamoxifen treatment. Although robust recombination of the *Rosa26^{LSL-ZsGreen}* allele was observed in the liver (*Figure 3A*), no ZsGreen was observed in the bone marrow (*Figure 3B*). Hepatocytes were the major cell type that expressed THPO in the liver after irradiation (*Figure 3C and D*). This finding suggests that hepatocytes are the major source of THPO with no appreciable THPO expression in the bone marrow following radioablation.

To functionally test whether osteoblasts and bone marrow mesenchymal cells are sources of THPO for hematopoietic recovery and HSC regeneration after irradiation, we gave *Col1a1-cre; Thpo^{fl/gfp}*, *Lepr^{cre}; Thpo^{fl/gfp}* or *Prrx1-cre; Thpo^{fl/gfp}* mice, along with littermate controls, a single 5.5 Gy dose of radiation and analyzed them 4 weeks later. Compared with irradiated controls, *Col1a1-cre; Thpo^{fl/gfp}*, *Lepr^{cre}; Thpo^{fl/gfp}* or *Prrx1-cre; Thpo^{fl/gfp}* mice showed no significant differences in leukocyte, neutrophil, and platelet counts, as well as reticulocyte frequency (*Figure 3E–H*). Bone marrow cellularity, LSK and HSC frequencies were also normal (*Figure 3I and J*, *Figure 3—figure supplement 2A*). These mice also had normal spleen cellularity, LSK and HSC frequencies, as well as total HSC numbers (*Figure 3K–M* and *Figure 3—figure supplement 2B*). These data suggest that osteoblasts and bone marrow mesenchymal stromal cells are dispensable sources of THPO for HSC and hematopoietic recovery from radioablative challenges.

To test the role of hepatocyte-derived THPO, we then irradiated *Thpo^{fl/fl}* mice treated with AAV8-TBG-cre virus vector. Compared to controls, these mice showed normal responses to irradiation in leukocyte, neutrophil, and platelet counts as well as reticulocyte frequency (*Figure 4—figure*

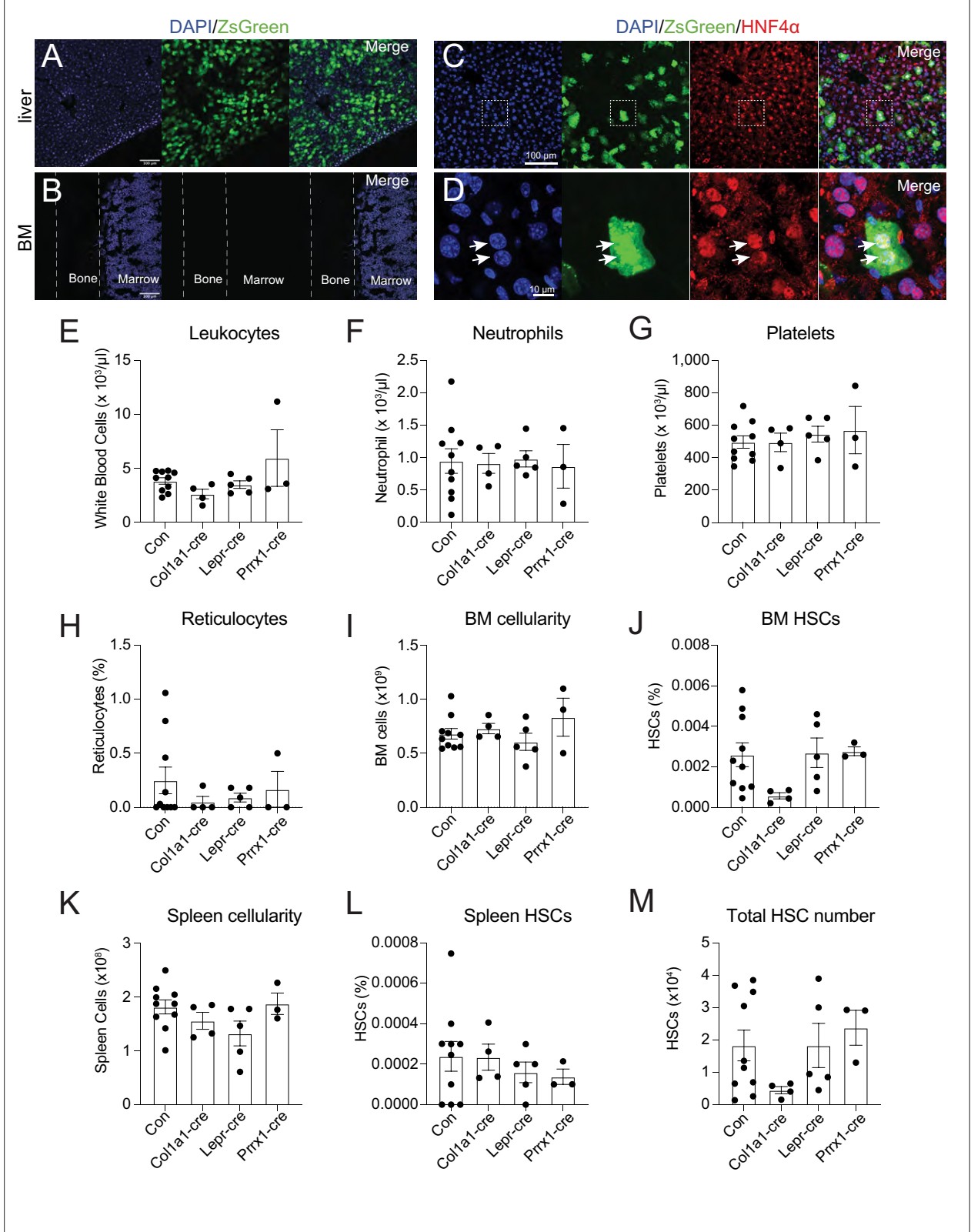

**Figure 3.** Bone marrow thrombopoietin (THPO) is not required for hematopoietic recovery from irradiation. (**A**) Confocal images of liver sections from tamoxifen-treated *Thpo*^*creER*^; *Rosa26*^*LSL-ZsGreen*^ mice 10 days after irradiation. (**B**) Confocal images of femur sections from tamoxifen-treated *Thpo*^*creER*^; *Rosa26*^*LSL-ZsGreen*^ mice 10 days after irradiation. (**C and D**) Confocal images showing immunostaining of HNF4α on liver sections from tamoxifen-treated *Thpo*^*creER*^; *Rosa26*^*LSL-ZsGreen*^ mice 10 days after irradiation. (**D**) Enlarged image of the region denoted in C. Arrows point to individual HNF4α⁺ hepatocytes.

*Figure 3 continued on next page*

*Figure 3 continued*

(**E–H**) Blood counts of mice with *Thpo* conditionally deleted from osteoblasts (Col1a1-cre), bone marrow stromal cells (Lepr-cre), or both (Prrx1-cre), and controls after irradiation. n = 3–10 mice. (**I–L**) Cellularity and frequencies of hematopoietic stem cells (HSCs) in the bone marrow and spleens from mice with *Thpo* conditionally deleted from osteoblasts, bone marrow stromal cells, or both, and controls after irradiation. n = 3–11 mice. (**M**) Total numbers of HSCs in the bone marrow and spleens from mice with *Thpo* conditionally deleted from osteoblasts, bone marrow stromal cells, or both, and controls after irradiation. n = 3–11 mice. Con: *Thpo*$^{gfp/+}$ control mice. Col1a1-cre: *Col1a1-cre; Thpo*$^{fl/gfp}$ mice. Lepr-cre: *Lepr*$^{cre}$*; Thpo*$^{fl/gfp}$ mice. Prrx1-cre: *Prrx1-cre; Thpo*$^{fl/gfp}$ mice. Data represent mean ± SEM. Statistical significance was assessed with one-way ANOVA with Dunnett's test. Each dot represents one independent mouse in E–M.

The online version of this article includes the following source data and figure supplement(s) for figure 3:

**Source data 1.** Numerical values of the data plotted in panels E-M.

**Figure supplement 1.** Hematopoietic recovery after irradiation depends on thrombopoietin (THPO).

**Figure supplement 1—source data 1.** Numerical values of the data plotted in panels A-N and P-W.

**Figure supplement 2.** Bone marrow thrombopoietin (THPO) is not required for hematopoietic recovery after irradiation.

**Figure supplement 2—source data 1.** Numerical values of the data plotted in panels A and B.

supplement 1A–H). Although the bone marrow cellularity response to irradiation was similar between *Thpo*$^{fl/fl}$ mice treated with AAV and control, there was a significant reduction in bone marrow HSC but not LSK frequency recovery in AAV-treated *Thpo*$^{fl/fl}$ mice relative to controls after irradiation (*Figure 4A–D* and *Figure 4—figure supplement 1I and J*). Compared with controls, spleen cellularity, LSK and HSC frequencies from *Thpo*$^{fl/fl}$ mice treated with AAV had normal responses after irradiation (*Figure 4E–H* and *Figure 4—figure supplement 1K and L*). Overall, *Thpo*$^{fl/fl}$ mice treated with AAV had a significant reduction in total HSC number recovery after irradiation (*Figure 4I and J*). Importantly, bone marrow cells from these irradiated AAV-treated mice showed functional defects in reconstituting irradiated recipients (*Figure 4K*). These data suggest that hepatic THPO is required for proper functional recovery of HSCs following irradiation-mediated myeloablation.

## Discussion

Our data show that the bone marrow does not meaningfully upregulate THPO protein after 5-FU treatment or irradiation. Although we did not exhaustively investigate all models of ablative conditioning, it appears that the bone marrow is not a functional source of THPO in hematopoietic recovery from acute myeloablative stress. However, our data do not rule out the possibility that bone marrow may serve as a source of THPO in certain specific stress conditions, such as idiopathic thrombocytopenia purpura (*Sungaran et al., 1997*). Future investigation into additional stress conditions is needed. We found that hepatocytes are an important source of THPO for stress hematopoiesis. Although several other studies have investigated the role of THPO on hematopoietic recovery after stress, the source of THPO and its impact on HSCs (rather than hematopoietic progenitors) have not been determined. By examining HSCs and hematopoietic progenitors, we show that HSCs are more sensitive to acute THPO perturbation than hematopoietic progenitors (LSKs), suggesting the major role of THPO in myeloablation is to promote HSC regeneration, at least under the conditions we studied. To our knowledge, THPO is the first systemic factor required for HSC regeneration and hematopoietic recovery identified to date. The presence of a stem cell hormone, like THPO, raises the possibility of other endocrine regulators of HSC regeneration and hematopoietic recovery.

Numerous connections between hepatic and hematopoietic pathophysiology have long been appreciated. Aplastic anemia is a known sequela of pediatric liver failure, and thrombocytopenia is a well-characterized feature of hepatic disease (*Hadzić et al., 2008*; *Peck-Radosavljevic, 2017*; *Tung et al., 2000*). While mechanisms such as metabolic toxicity and splenic sequestration may play a role in mediating these effects, it seems likely that disruption of the THPO signaling is also involved. Indeed, several recent clinical studies have shown that newly developed THPO agonists can reduce the need for platelet transfusions in patients with chronic liver disease (*Peck-Radosavljevic et al., 2019*; *Xu and Cai, 2019*). While early trials of THPO mimetics in primary and acquired aplastic anemia have been promising, to our knowledge, no group has

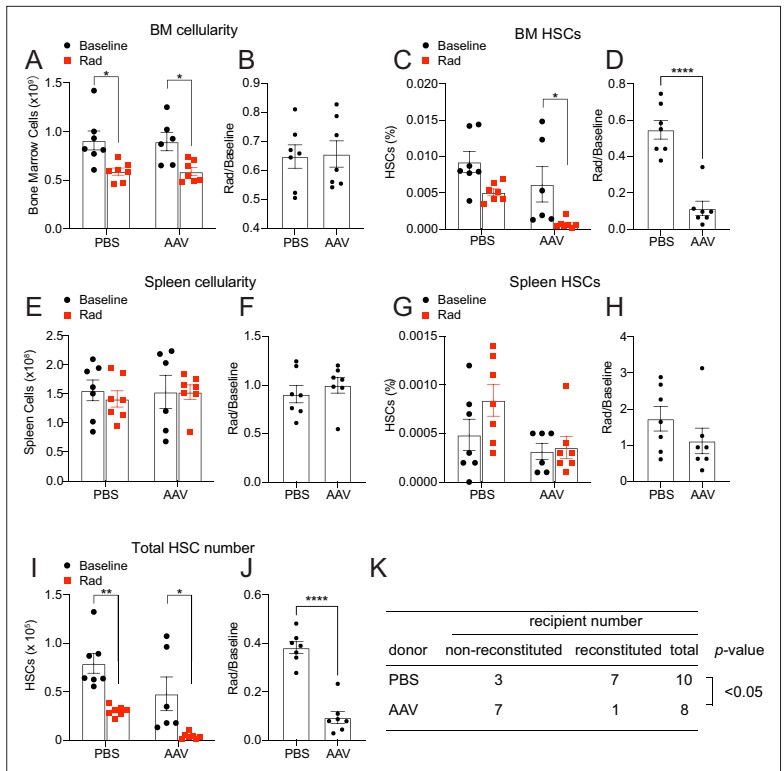

**Figure 4.** Hepatic thrombopoietin (THPO) is required for effective hematopoietic stem cell (HSC) recovery from ionizing radiation. (**A–H**) Cellularity and frequencies of HSCs in the bone marrow and spleens from mice with *Thpo* conditionally deleted from hepatocytes (AAV) and controls with and without irradiation. Normalized fold changes after irradiation (relative to no irradiation baseline) are also shown. n = 6–7 mice. (**I and J**) Total numbers of HSCs in the bone marrow and spleens from mice with *Thpo* conditionally deleted from hepatocytes (AAV) and controls with and without irradiation. Normalized fold changes after irradiation (relative to no irradiation baseline) are also shown. n = 6–7 mice. (**K**) Numbers of total recipients and long-term multilineage reconstituted recipients after transplantation of 1,000,000 bone marrow cells from PBS- or AAV-treated *Thpo^{fl/fl}* and irradiated donor mice along with 1,000,000 irradiated competitor bone marrow cells. Recipients were scored as reconstituted if they showed donor-derived myeloid, B and T cells in the peripheral blood 16 weeks after transplantation. Data were from recipients of two donor pairs from two independent experiments. PBS: PBS-treated *Thpo^{fl/fl}* or wild-type control mice. AAV: AAV8-TBG-cre-treated *Thpo^{fl/fl}* mice. Data represent mean ± SEM. *p < 0.05, **p < 0.01, ****p < 0.0001. Statistical significance was assessed with two-way ANOVA with Turkey's test (**A, C, E, G, I**), Student's t-test (**B, D, F, H, J**), or Fisher's exact test (**K**). Each dot represents one independent mouse in A–J.

The online version of this article includes the following source data and figure supplement(s) for figure 4:

**Source data 1.** Numerical values of the data plotted in panels A-J.

**Figure supplement 1.** Hepatic thrombopoietin (THPO) is critical for hematopoietic recovery after irradiation.

**Figure supplement 1—source data 1.** Numerical values of the data plotted in panels A-L.

specifically studied their use in liver failure-associated aplastic anemia (*Desmond et al., 2014*; *Gill et al., 2017*). Given that hepatic *Thpo* mRNA and serum THPO levels are significantly down-regulated in liver failure (*Wolber et al., 1999*), further studies are warranted. Our data suggest that cancer patients with underlying liver disease may have greater sensitivity to myeloablative conditioning and supplementing THPO mimetics may help alleviate the adverse effects associated with myeloablative conditioning in these patients. On the other hand, THPO may also have effects on cells beyond the hematopoietic system. For example, a recent report suggests that in hepatocellular carcinoma, THPO may facilitate tumor progression by promoting VEGF signaling in hepatocytes (*Vizio et al., 2021*). Thus, strategies aimed at manipulating the THPO pathway need careful evaluation.

# Materials and methods

**Key resources table**

| Reagent type (species) or resource | Designation | Source or reference | Identifiers | Additional information |
|---|---|---|---|---|
| Genetic reagent (*Mus musculus*) | *Prrx1-cre* | PMID:12112875 | | JAX stock (005584) |
| Genetic reagent (*Mus musculus*) | *Lepr^{cre}* | PMID:11283374 | | JAX stock (008320) |
| Genetic reagent (*Mus musculus*) | *Rosa26^{LSL-ZsGreen}* | PMID:20023653 | | JAX stock (007906) |
| Genetic reagent (*Mus musculus*) | *Col1a1-cre* | PMID:15470637 | | |
| Genetic reagent (*Mus musculus*) | *Thpo^{gfp}* | PMID:29622652 | | |
| Genetic reagent (*Mus musculus*) | *Thpo^{creER}* | PMID:29622652 | | |
| Genetic reagent (*Mus musculus*) | *Thpo^{fl}* | PMID:29622652 | | |
| Recombinant DNA reagent | AAV8-TBG-cre | Penn Vector core or Addgene | Cat# 107787-AAV8 | |
| Chemical compound, drug | 5-Fluorouracil | Fresenius Kabi | Cat# 101,710 | 150 mg/kg IP |
| Chemical compound, drug | Tamoxifen | Sigma | Cat# T5648 | |
| Chemical compound, drug | Collagenase, Type IV | Worthington | Cat# LS004188 | |
| Chemical compound, drug | DNase I | Sigma | Cat# D4527 | |
| Antibody | (Rat monoclonal) anti-CD2 (RM2-5) | Biolegend | | Flow cytometry (1:200) |
| Antibody | (Rat monoclonal) anti-CD3 (17A2) | Biolegend | | Flow cytometry (1:200) |
| Antibody | (Rat monoclonal) anti-CD5 (53–7.3) | Biolegend | | Flow cytometry (1:400) |
| Antibody | (Rat monoclonal) anti-CD8a (53–6.7) | Biolegend | | Flow cytometry (1:400) |
| Antibody | (Rat monoclonal) anti-B220 (6B2) | Biolegend | | Flow cytometry (1:400) |
| Antibody | (Rat monoclonal) anti-Gr1 (8C5) | Biolegend | | Flow cytometry (1:400) |
| Antibody | (Rat monoclonal) anti-Ter119 | Biolegend | | Flow cytometry (1:200) |
| Antibody | (Rat monoclonal) anti-Sca1 (E13-161.7) | Biolegend | | Flow cytometry (1:200) |
| Antibody | (Rat monoclonal) anti-cKit (2B8) | Biolegend | | Flow cytometry (1:200) |
| Antibody | Armenian (Hamster monoclonal) anti-CD48 (HM48-1) | Biolegend | | Flow cytometry (1:200) |
| Antibody | (Rat monoclonal) anti-CD150 (TC15-12F12.2) | Biolegend | | Flow cytometry (1:200) |

*Continued on next page*

*Continued*

| Reagent type (species) or resource | Designation | Source or reference | Identifiers | Additional information |
|---|---|---|---|---|
| Antibody | (Rat monoclonal) anti-CD45.2 (104) | Biolegend | | Flow cytometry (1:400) |
| Antibody | (Rat monoclonal) anti-CD45.1 (A20) | Biolegend | | Flow cytometry (1:400) |
| Antibody | (Rat monoclonal) anti-Mac1 (M1/70) | Biolegend | | Flow cytometry (1:400) |
| Antibody | (Rat monoclonal) anti-CD45 (30 F-11) | Biolegend | | Flow cytometry (1:400) |
| Antibody | (Rabbit monoclonal) anti-HNF4α (EPR16885-99) | AbCam | Cat# Ab201460 | IF (1:10) |
| Sequence-based reagent | OLD815 | IDT DNA | CCACCACCATGCCTAACTCT | |
| Sequence-based reagent | OLD816 | IDT DNA | GTTCTCCTCCACGTCTCCAG | |
| Sequence-based reagent | OLD817 | IDT DNA | TCGCTAGCTGCTCTGATGAA | |
| Sequence-based reagent | ZsGreen F | IDT DNA | GGCATTAAAGCAGCGTATCC | |
| Sequence-based reagent | ZsGreen R | IDT DNA | AACCAGAAGTGGCACCTGAC | |
| Sequence-based reagent | OLD581 | IDT DNA | CATCTCGCTGCTCTTAGCAGGG | |
| Sequence-based reagent | OLD582 | IDT DNA | GAGCTGTTTGTGTTCCAACTGG | |
| Sequence-based reagent | OLD292 | IDT DNA | CGGACACGCTGAACTTGTGG | |
| Sequence-based reagent | OLD528 | IDT DNA | ACTTATTCTCAGGTGGTGACTC | |
| Sequence-based reagent | OLD653 | IDT DNA | AGGGAGCCACTTCAGTTAGAC | |
| Sequence-based reagent | OLD434 | IDT DNA | CATTGTATGGGATCTGATCTGG | |
| Sequence-based reagent | OLD435 | IDT DNA | GGCAAATTTTGGTGTACGGTC | |
| Sequence-based reagent | OLD338 | IDT DNA | GCATTTCTGGGGATTGCTTA | |
| Sequence-based reagent | OLD339 | IDT DNA | ATTCTCCCACCGTCAGTACG | |
| Sequence-based reagent | OLD390 | IDT DNA | CCTTTGTCTATCCCTGTTCTGC | |
| Sequence-based reagent | OLD391 | IDT DNA | ACTGCCCCTAGAATGTCCTGT | |
| Sequence-based reagent | OLD27 | IDT DNA | GCTCTTTTCCAGCCTTCCTT | |
| Sequence-based reagent | OLD28 | IDT DNA | CTTCTGCATCCTGTCAGCAA | |
| Software, algorithm | FacsDiva | BD | | |
| Software, algorithm | FlowJo | FlowJo | | |
| Software, algorithm | Prism | GraphPad | | |
| Other | SuperScript III | ThermoFisher | Cat# 18080093 | |
| Other | ProtoScript II | NEB | Cat# M0368S | |
| Other | PELCO Cryo-Embedding compound | Ted Pella, Inc | Cat# 27,300 | |

## Mice

All mice were 8–16 weeks of age and maintained on a C57BL/6 background. *Prrx1-cre* (stock # 005584) recombining in bone marrow mesenchymal cells of the long bones, *Lepr^cre^* (stock # 008320) recombining in bone marrow mesenchymal stromal cells, and *Rosa26^LSL-ZsGreen^* (stock # 007906) mice were obtained from the Jackson Laboratory. *Col1a1-cre* mice (with 2.3 kb promoter), recombining in mature osteoblasts, were described previously (***Liu et al., 2004***). The generation of *Thpo^gfp^*, *Thpo-^creER^*, and *Thpo^fl^* mice was described previously (***Decker et al., 2018***). *Thpo^gfp^* is a null allele of *Thpo*. *Thpo^creER^* knockin mice allows tracing of cells that translate THPO protein. All mice were housed in

specific pathogen-free, Association for the Assessment and Accreditation of Laboratory Animal Care (AAALAC)-approved facilities at the Columbia University Medical Center. All protocols were approved by the Institute Animal Care and Use Committee of Columbia University.

## Genotyping PCR

Primers for genotyping $Thpo^{creER}$: OLD815, 5'-CCACCACCATGCCTAACTCT-3'; OLD816, 5'-GTTCTCCTCCACGTCTCCAG-3'; and OLD817, 5'-TCGCTAGCTGCTCTGATGAA-3'.

Primers for genotyping $Rosa26^{LSL-ZsGreen}$: GGCATTAAAGCAGCGTATCC and AACCAGAAGTGGCACCTGAC.

Primers for genotyping $Thpo^{fl}$: OLD581, 5'-CATCTCGCTGCTCTTAGCAGGG-3' and OLD582, 5'-GAGCTGTTTGTGTTCCAACTGG-3'.

Primers for genotyping $Thpo^{gfp}$: OLD292, 5'-CGGACACGCTGAACTTGTGG-3'; OLD528 5'-ACTTATTCTCAGGTGGTGACTC-3' and OLD653 5'-AGGGAGCCACTTCAGTTAGAC-3'.

Primers for genotyping $Lepr^{cre}$: OLD434 5'-CATTGTATGGGATCTGATCTGG-3' and OLD435 5'-GGCAAATTTTGGTGTACGGTC-3'.

Primers for genotyping $cre$: OLD338, 5'-GCATTTCTGGGGATTGCTTA-3' and OLD339, 5'-ATTCTCCCACCGTCAGTACG-3'.

## Chemoablative challenge

5-FU (150 mg/kg) was administered to mice via a single intraperitoneal injection. Ten days post-injection, mice were euthanized for analysis.

## Radioablative challenge

Mice were irradiated by a Cesium 137 Irradiator (JL Shepherd and Associates) at 300 rad/min with one dose of 550 rad. Four weeks after irradiation, mice were euthanized for analysis.

## Tamoxifen administration

Tamoxifen (Sigma) was dissolved in corn oil for a final concentration of 20 mg/mL. Every other day for 10 days, 50 µL of the solution was administered by oral gavage. Mice were analyzed 2–4 days after the final tamoxifen administration.

## Viral vector infections

Replication-incompetent AAV8-TBG-cre was obtained from the Penn Vector Core or Addgene. AAV8-TBG-cre vector carries $cre$ recombinase gene under the regulatory control of hepatocyte-specific thyroid-binding globulin (TBG) promoter. Efficient recombination was achieved at a dose of $2.5 \times 10^{11}$ viral particles diluted in sterile $1 \times$ PBS. AAV8-TBG-cre was administered to mice via retro-orbital venous sinus injection.

## Complete blood count

Peripheral blood was collected to EDTA tubes (Microvette CB300, Sarstedt) to prevent clogging. Blood samples were then run on a complete blood count analyzer (Genesis, Oxford Science).

## Flow cytometry

Bone marrow cells were isolated by flushing the long bones or by crushing the long bones, pelvis, and vertebrae with mortar and pestle in $Ca^{2+}$- and $Mg^{2+}$-free HBSS with 2 % heat-inactivated bovine serum. Spleen cells were obtained by crushing the spleens between two glass slides. The cells were passed through a 25 G needle several times and filtered with a 70 µm nylon mesh. The following antibodies were used to perform HSC staining: lineage markers (anti-CD2 [RM2-5], anti-CD3 [17A2], anti-CD5 [53–7.3], anti-CD8a [53–6.7], anti-B220 [6B2], anti-Gr-1 [8C5], anti-Ter119), anti-Sca-1 (E13-161.7), anti-c-kit (2B8), anti-CD48 (HM48-1), anti-CD150 (TC15-12F12.2).

For flow cytometric sorting of bone marrow stromal cells, bone marrow plugs were flushed as described above, then digested with collagenase IV (200 U/mL) and DNase1 (200 U/mL) at 37 °C for 20 min. Samples were then stained with anti-CD45 (30 F-11) and anti-Ter119 antibodies and sorted on a flow cytometer.

## Quantitative real-time PCR

Cells were lysed in Trizol. Total RNA was extracted according to manufacturer's instructions. Total RNA was subjected to reverse transcription using SuperScript III (Invitrogen) or ProtoScript II (NEB). Quantitative real-time PCR was run using SYBR green on a CFX Connect system (Biorad). β-Actin (*Actb*) was used to normalize the RNA content of samples. Primers used were: *Thpo*: OLD390, CCTTTGTCTATCCCTGTTCTGC and OLD391, ACTGCCCCTAGAATGTCCTGT; *β-actin*: OLD27, 5′-GCTCTTTTCCAGCCTTCCTT-3′ and OLD28, 5′-CTTCTGCATCCTGTCAGCAA-3′.

## Bone and liver sectioning

Freshly disassociated long bones were fixed for 3 hr in a solution of 4 % paraformaldehyde, 7 % picric acid, and 10 % sucrose (W/V). The bones were then embedded in 8 % gelatin, immediately snap-frozen in liquid $N_2$, and stored at –80 °C. Bones were sectioned using a CryoJane system (Instrumedics). For liver, cardiac perfusion with formalin was performed immediately after mouse sacrifice, and perfused liver tissue was dehydrated in a 30 % sucrose solution overnight at 4 °C. Liver tissue was then placed in PELCO Cryo-Embedding compound (Ted Pella, Inc), frozen on dry ice, and stored at –80 °C. Liver tissue was sectioned and directly transferred onto microscope slides. Both bone and liver sections were dried overnight at room temperature and stored at –80 °C. Sections were rehydrated in PBS for 5 min, stained with DAPI for 15 min, then mounted with Vectashield Hardset Antifade Mounting Medium (Vector Laboratories). An rabbit anti-HNF4α antibody (AbCam, ab201460) was used to stain hepatocytes. Images were acquired on a Nikon Eclipse Ti confocal microscope (Nikon Instruments) or a Leica SP8 confocal microscope (Leica Microsystems).

## Long-term competitive reconstitution assay

Adult recipient mice were lethally irradiated by a Cesium 137 Irradiator (JL Shepherd and Associates) at 300 rad/min with two doses of 550 rad (total 1100 rad) delivered at least 2 hr apart. Cells were transplanted by retro-orbital venous sinus injection of anesthetized mice. Donor bone marrow cells were transplanted along with recipient bone marrow cells into lethally irradiated recipient mice. For irradiation-induced myeloablation, the competing recipient bone marrow cells were from mice that were irradiated similarly as the donor mice. Mice were maintained on antibiotic water (Baytril 0.17 g/L) for 14 days, then switched to regular water. Recipient mice were periodically bled to assess the level of donor-derived blood lineages, including myeloid, B, and T cells for at least 16 weeks. Blood was subjected to ammonium chloride potassium red cell lysis before antibody staining. Antibodies including anti-CD45.2 (104), anti-CD45.1 (A20), anti-CD3 (17A2), anti-B220 (6B2), anti-Gr-1 (8C5), and anti-Mac-1 (M1/70) were used to stain cells. Mice with presence of donor-derived myeloid, B, and T cells for 16 weeks were considered as long-term multilineage reconstituted.

## Statistical analysis

All analyses were done using GraphPad Prism 7.0. In any comparison between a pooled control cohort and multiple experimental conditions, we used one-way ANOVA with Dunnett's test. In any comparison between control and mutant in multiple experimental conditions, we used two-way ANOVA with Turkey's test. For surviving curve comparison, we used Log-rank (Mantel-Cox) test. For transplantation experiments, we used Fisher's exact test. For all other comparisons, we used unpaired t-test. In all figures, error bars represent standard error of mean.

## Acknowledgements

This work was supported by the National Heart, Lung and Blood Institute (R01HL132074). LG was supported by the NYSTEM Columbia training program in stem cell research and a Columbia Stem Cell Initiative Seed Grant. MD was supported by the Columbia Medical Scientist Training Program and the NIH (1F30HL137323). LD was supported by the Rita Allen Foundation, the Irma Hirschl Research Award, the Schaefer Research Scholar Program, and the Leukemia and Lymphoma Society Scholar award, R01HL153487 and R01HL155868. Images were collected in the Confocal and Specialized Microscopy Shared Resource of the Herbert Irving Comprehensive Cancer Center at Columbia University, supported by NIH grant P30CA013696 (National Cancer Institute). We thank M Kissner at the Columbia Stem Cell Initiative for help on flow cytometry.

## Additional information

### Funding

| Funder | Grant reference number | Author |
|---|---|---|
| National Heart, Lung, and Blood Institute | R01HL132074 | Lei Ding |
| National Heart, Lung, and Blood Institute | R01HL153487 | Lei Ding |
| National Heart, Lung, and Blood Institute | R01HL155868 | Lei Ding |
| New York State Stem Cell Science | Training grant | Longfei Gao |
| National Heart, Lung, and Blood Institute | 1F30HL137323 | Matthew Decker |
| Rita Allen Foundation | Scholar Award | Lei Ding |
| Irma T. Hirschl Trust | Research Awards | Lei Ding |
| Leukemia and Lymphoma Society | Scholar Award | Lei Ding |
| National Cancer Institute | P30CA013696 | Longfei Gao Matthew Decker Haidee Chen Lei Ding |

The funders had no role in study design, data collection and interpretation, or the decision to submit the work for publication.

### Author contributions

Longfei Gao, Data curation, Formal analysis, Investigation, Writing - review and editing; Matthew Decker, Data curation, Formal analysis, Investigation, Writing - original draft, Writing - review and editing; Haidee Chen, Investigation, Project administration; Lei Ding, Conceptualization, Funding acquisition, Project administration, Supervision, Writing - original draft, Writing - review and editing

### Author ORCIDs

Longfei Gao http://orcid.org/0000-0002-2331-8471
Lei Ding http://orcid.org/0000-0003-4869-8877

### Ethics

All mice were housed in specific pathogen-free, Association for the Assessment and Accreditation of Laboratory Animal Care (AAALAC)- approved facilities at the Columbia University Medical Center. All protocols were approved by the Institute Animal Care and Use Committee of Columbia University under AC-AAAZ9451.

### Decision letter and Author response

Decision letter https://doi.org/10.7554/eLife.69894.sa1
Author response https://doi.org/10.7554/eLife.69894.sa2

## Additional files

### Supplementary files
• Transparent reporting form

### Data availability

All data were presented with individual data points from each mouse. Source data files have been provided. No sequencing or diffraction data are generated.

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
