## [Decision Letter]

**Acceptance summary:**

This paper shows that hepatic TPO is responsible for HSC regeneration after 5FU and irradiation. The paper is novel and will shift the paradigm in the field. The studies are conducted carefully and the experimental approach is sound. The authors have been responsive to the previous comments and have modified the manuscript significantly.

**Decision letter after peer review:**

Thank you for submitting your article "Thrombopoietin from hepatocytes promotes hematopoietic stem cell regeneration after myeloablation" for consideration by *eLife*. Your article has been reviewed by 3 peer reviewers, including Hossein Ardehali as the Reviewing Editor and Reviewer #1, and the evaluation has been overseen by a Reviewing Editor and Mone Zaidi as the Senior Editor.

Essential revisions:

1) The authors only assessed the transcriptional regulation of TPO after 5FU in the bone marrow. They need to also assess the transcript level in the liver.

2) What is the role of TPO expression from osteoblasts and mesenchymal stem cells if it is not involved in HSC regeneration after myeloablation?

3) Although the studies are focused on the number of hematopoietic cells after 5-FU and irradiation, it would be helpful to see if the difference in the numbers of these cells is associated with any disorders, like increased inflammation or bleeding. Also, it would be helpful to determine whether deletion of TPO in the liver is associated with anemia in mice.

4) What it the rational to move some of the data to the supplements, while others are kept in the main figures? e.g Figure 1 showing no effects in the deleted mice throughout the figure, in Figure 2 the leukocytes, neutrophiles counts etc. moved to S3?

5) AAV stands for adeno-associated viral vector, not adenovirus and also not adeno-associated virus, as the AAV used in the study is replication incompetent and should be addressed as vector.

6) How was the reticulocyte count determined?

7) There are various Lepr-Cre-deleter mice at Jackson lab (same for the other lines). It is not sufficient to name them Lepr-Cre or Prx1-Cre in the Materials and methods part, please give the correct name of the mouse line.

8) Figure legends for the supplementary Figures are missing.

9) In respect to the discussion: The systemic effect of Thpo regulation has been discussed by deGraaf et al., 2010.

10) It would be of advantage for the reader, to mention and explain the results from their previous study (the tissue-specific deletion of Thpo under steady state conditions in the mouse) in the introduction.

11) Platelet counts in PBS treated mice (Figure 2A) are lower than in untreated control mice (Figure 1G). What is the major difference between these controls?

12) The paper would benefit to directly compare the data from the osteoblast and stromal cell deleted mice with the AAV treated mice in one graphic. Except for the difference in technology used to delete Thpo there is no obvious reason why the data are split up in the different figures.

13) The response in platelet counts after 5-FU administration in Mpl-/- mice has been studied (Levin et al., Blood 2001). These mice can respond in the absence of intact Thpo/Mpl signaling. These data are contradictory to what is shown here. What could be the explanation for that?

14) The major difference between using Cre-deleter mice in contrast to the AAV mediated application is that in the latter case Thpo is deleted in an adult mouse while in the other case, mice have develop under the condition that no Thpo is supplied from the bone marrow niche (osteoblasts or mesenchymal cells, accordingly). This questions whether these two scenarios can be safely compared. Similar to this, the inhibition of Thpo/Mpl signaling by a dominant-negative Mpl receptor in the adult mouse causes severe and lethal aplasia (Wicke et al., Mol Ther 2010), while the germ line knockout of Mpl or Thpo is in general well tolerated in the mouse and HSC defects get only more obvious after transplantation (Alexander et al., Blood 1996; Kimura et al., PNAS 1998; deSauvage et al., J Ex Med 1996; Yoshihara et al., CSC 2007, Qian et al., CSC 2007). What was the rational to use the AAV8 Cre delivery in contrast to using a deleter mouse with liver-specific Cre expression?

15) The spread in the data after AAV application to mice may be caused by differences in the transduction efficiencies in the liver. How efficient was the deletion in the liver after AAV application?

16) In the transplantation experiment, the donor mice were treated with AAV-Cre. To address the role of hepatic Thpo in the recovery after transplantation, it would be of interest to delete Thpo in the recipients.

17) The transplantation of irradiated BM cells should in general not lead to a successful engraftment of these cells. What is the aim of this experiment?

18) What is the threshold of donor contribution that is judged as positive engraftment?

19) As mentioned above, the study shows that, similarly to what occurs in bone marrow HSC maintenance in steady-state conditions, hepatic TPO appears to be a major functional trigger for hematopoietic recovery and HSC regeneration after myelosuppression caused by radio-chemotherapy injury.

To avoid misleading conclusions and thus contribute to clinical medicine more precisely, the experimental design of the study must relate to a tumor context, since radio-chemotherapies play an established role in the treatment of cancer patients. The genetically modified animal models must have pathophysiological characteristics as comparable as possible to those that occur in humans (for animal-to-human extrapolation).

We thus recommend that the reported experiments be flanked by others run upon a murine tumor model.

20) Since it has been shown that TPO is an important and nonredundant regulator of VEGF-A production in HSCs, and that is activates a VEGF-A internal autocrine loop, participating in the favorable TPO effects on HSC self-renewal and expansion (Kirito K et al., Blood. 2005; 105:4258) it would be interesting to investigate the contribution of this mechanism to the efficacy of hepatic TPO in hematopoietic recovery in stress conditions in genetically altered mouse models. This would be of interest in particular because VEGF-A, notoriously elevated in a tumor context, could also be counted among the endocrine factors which, together with TPO, contribute to bone marrow recovery, as suggested in the discussion.

21) In the discussion, in the light of their data, the authors assume that cancer patients with underlying liver disease, such as HCC, may have enhanced sensitivity to myeloablative conditioning because of liver-failure-related hepatic downregulated TPO levels. Since TPO mimetic drugs, such as romiplostim and eltrombopag, have been shown to improve recovery after ablative challenge, the authors suggest that these agents could help alleviate the adverse effects associated with myeloablative conditioning in such patients.

However, it has recently been reported that, in the HCC context (Vizio B et al., Int. J. Mol. Sci. 2021, 22, 1818), the increased hepatic TPO level appears to be a novel triggering signal to promote tumor progression. Therefore, curative strategies aimed at manipulating THPOR in HCC patients should be evaluated with extreme care.

The possibility of tumor conditioning should be pointed out, and recent literature in this connection included in the discussion.

---

## [Author Response]

Essential revisions:1) The authors only assessed the transcriptional regulation of TPO after 5FU in the bone marrow. They need to also assess the transcript level in the liver.

It has been reported that *Thpo* transcripts in the liver are expressed at constant

levels (Blood 1995 86:3668). Because protein production is the ultimate readout of THPO production and *Thpo* is under profound translational control (Blood 1998 92:4023), we focused on assessing THPO translation using our translational reporter mice. Nonetheless, per the request by the reviewer, we performed qPCR analysis (see Author response image 1). Consistent with the prior report, *Thpo* transcript levels were not changed in the liver after 5-FU treatment.

**Author response image 1. sa2fig1:** 5-FU treatment does not change *Thpo* transcript levels in the liver. qPCR analysis showing the relative expression levels of *Thpo* transcripts in controls and 5-FU-treated mice. n = 4 mice for each condition.

2) What is the role of TPO expression from osteoblasts and mesenchymal stem cells if it is not involved in HSC regeneration after myeloablation?

This is a great question that intrigues us as well. We initially hypothesized that THPO from the local bone marrow niche is for HSC regeneration after myeloablation, but this does not turn out to be true. Because only 5-FU and irradiation conditions were tested, we can not completely rule out the possibility that locally produced THPO plays roles for HSC regeneration in other stress conditions. We have addressed this in the Discussion section.

3) Although the studies are focused on the number of hematopoietic cells after 5-FU and irradiation, it would be helpful to see if the difference in the numbers of these cells is associated with any disorders, like increased inflammation or bleeding. Also, it would be helpful to determine whether deletion of TPO in the liver is associated with anemia in mice.

Thank you for raising these points. Inflammation or bleeding is not considered as myeloablation. However, as the reviewer pointed out, inflammation and bleeding are also interesting conditions to look at. This is something we plan to work on in the future and out of the scope of the current study. Please note that, regardless what the results of these future experiments may be, they will not change the main conclusion of the current manuscript, which focuses on the major acute myeloablation conditions. The impact of liver-derived THPO on reticulocytes depends on the treatment condition. Under the steady-state condition, deletion of *Thpo* from the liver did not lead to a significant alteration of reticulocyte frequencies (Figure 2-figure supplement 1 H, I and Figure 4—figure supplement 1 G, H). Under 5-FU treatment, deletion of *Thpo* from the liver led to a trend in reduction of reticulocyte frequency compared with controls (Figure 2—figure supplement 1 H, I). Under irradiation condition, deletion of *Thpo* from the liver did not lead to a significant difference of reticulocyte frequencies (Figure 4—figure supplement 1 G, H).

4) What it the rational to move some of the data to the supplements, while others are kept in the main figures? e.g Figure 1 showing no effects in the deleted mice throughout the figure, in Figure 2 the leukocytes, neutrophiles counts etc. moved to S3?

We apologize for the confusion. We organized the data based on myeloablative conditions (5-FU vs irradiation) and where *Thpo* is deleted (local bone marrow vs the liver) in the original version of the manuscript. Because deletion of *Thpo* from the liver leads to phenotypes in the hematopoietic system under the steady-state condition, we need to show normalized phenotypes, such as in Figure 2A. As a result, the data from mice with *Thpo* deleted from the liver are more complicated. Thus we moved the data showing no differences to Supplementary Figures in the original version of the manuscript. We agree with the reviewer that the data in the figures can be better organized to be consistent throughout. We modified the figures by moving all peripheral blood data in Figures 2 and 4 into Supplementary Figures in the revised manscript, as suggested by the reviewer.

5) AAV stands for adeno-associated viral vector, not adenovirus and also not adeno-associated virus, as the AAV used in the study is replication incompetent and should be addressed as vector.

Thank you for pointing out the confusion. We agree with reviewer and have made this clear by modifying the text to ‘replication incompetent hepatotropic AAV8-TBG-cre viral vector (AAV)’ in the revised manuscript.

6) How was the reticulocyte count determined?

The reticulocyte percentages were determined by a Complete Blood Count analyzer (Genesis, Oxford Science). We have added these details to the method section in the revised

manuscript.

7) There are various Lepr-Cre-deleter mice at Jackson lab (same for the other lines). It is not sufficient to name them Lepr-Cre or Prx1-Cre in the Materials and methods part, please give the correct name of the mouse line.

We have added the Jackson Lab stock number information in the Materials and methods section in the revised manuscript.

8) Figure legends for the supplementary Figures are missing.

We sincerely apologize for this omission. The legends for the supplementary figures have been added in the revised manuscript.

9) In respect to the discussion: The systemic effect of Thpo regulation has been discussed by deGraaf et al., 2010.

We have cited the paper and added discussion in the introduction section of the revised manuscript.

10) It would be of advantage for the reader, to mention and explain the results from their previous study (the tissue-specific deletion of Thpo under steady state conditions in the mouse) in the introduction.

We have added this information to the introduction section of the revised manuscript.

11) Platelet counts in PBS treated mice (Figure 2A) are lower than in untreated control mice (Figure 1G). What is the major difference between these controls?

Controls in Figure 1G are genotyping negative control mice treated with 5-FU.

Controls in the original Figure 2A (new Figure 2—figure supplement 1 F) are wild-type mice treated with PBS. 5-FU treatment increases platelet counts.

12) The paper would benefit to directly compare the data from the osteoblast and stromal cell deleted mice with the AAV treated mice in one graphic. Except for the difference in technology used to delete Thpo there is no obvious reason why the data are split up in the different figures.

We have considered the possibility of a direct comparison. Mice with *Thpo* deleted from the bone marrow did not have any phenotypes under the steady-state condition. A direct comparison among these mice under 5-FU or irradiation conditions would be appropriate (e.g. Figure 1). However, mice with *Thpo* deleted from the hepatocytes (AAV-treated mice) have hematopoietic phenotypes under the steady-state condition. We need to normalize the data under 5-FU and irradiation conditions to the baseline (mice with same genotype under the steady-state condition). It would be inappropriate and confusing to directly compare the data from the mice with *Thpo* deleted from the bone marrow with those with *Thpo* deleted from the liver (AAV-treated mice). Therefore, we separated the data of mice with *Thpo* deleted from the bone marrow from the data of mice with *Thpo* deleted from the liver (AAV-treated mice) into different figures.

13) The response in platelet counts after 5-FU administration in Mpl-/- mice has been studied (Levin et al., Blood 2001). These mice can respond in the absence of intact Thpo/Mpl signaling. These data are contradictory to what is shown here. What could be the explanation for that?

Thank you for raising this point. 5-FU induces the strongest platelet rebound at around 10 days after its administration (Experimental Hematology 2013 41:635, Blood 1992 80:904). We thus analyzed mice at day 10 after 5-FU administration. Note that Levin et al., (Blood 2001 98:1019) show that at day 10 after 5-FU treatment, *Mpl*-/- mice have significant defects in platelet recovery (see Figure 1A therein). At day 20-25, *Mpl*-/- mice can catch up and have relatively normal platelet counts compared to controls. Subsequently, their platelet levels decrease again. It appears that *Mpl*-/- mice can transiently produce platelet in respond to 5-FU through THPO-independent mechanisms, although the peak response is much lower and later than wild-type mice. Nonetheless, their day 10 data are consistent with ours in that efficient platelet rebound at this stage is *Thpo/Mpl*-dependent.

14) The major difference between using Cre-deleter mice in contrast to the AAV mediated application is that in the latter case Thpo is deleted in an adult mouse while in the other case, mice have develop under the condition that no Thpo is supplied from the bone marrow niche (osteoblasts or mesenchymal cells, accordingly). This questions whether these two scenarios can be safely compared. Similar to this, the inhibition of Thpo/Mpl signaling by a dominant-negative Mpl receptor in the adult mouse causes severe and lethal aplasia (Wicke et al., Mol Ther 2010), while the germ line knockout of Mpl or Thpo is in general well tolerated in the mouse and HSC defects get only more obvious after transplantation (Alexander et al., Blood 1996; Kimura et al., PNAS 1998; deSauvage et al., J Ex Med 1996; Yoshihara et al., CSC 2007, Qian et al., CSC 2007). What was the rational to use the AAV8 Cre delivery in contrast to using a deleter mouse with liver-specific Cre expression?

Thank you for raising this important point. Wicke et al., (Mol Ther 2010 18:343) used an *Mpl* overexpression model but not a dominant-negative *Mpl* model in adult mice. The adult inducible *Thpo* knockout mice have similar phenotypes in the hematopoietic system to the germline *Thpo* knockout mice (Science 2018 360:106), suggesting the adaptation to *Thpo* depletion during development is unlikely. However, this important point needs some consideration. No THPO protein (as can be detected by our reporter mice) is generated in the bone marrow. And mice with *Thpo* deleted from the bone marrow niche have no phenotypes, either under the steady-state condition or after myeloablation. It is thus unlikely that HSCs and the hematopoietic system adapt to *Thpo* deletion in these knockout mice. The most plausible explanation for the lack of phenotype is that bone marrow-derived THPO (if any) is not required for HSC regeneration after myeloablation. We have used mouse models with Cre specifically expressed in the hepatocytes (*Alb-cre*) to delete *Thpo*. *Alb-cre; Thpo*^*fl/fl*^ mice have a severe depletion of HSCs (virtually complete depletion) under the steady-state condition (Science 2018 360:106). It is thus technically difficult to normalize the parameters in myeloablation conditions to the baseline levels which have nearly no HSCs in these mice. In addition, *Alb-cre; Thpo*^*fl/fl*^ mice do not survive the myeloablation well. Thus, we used acute inducible *Thpo* deletion model with AAV8-TBG-cre.

15) The spread in the data after AAV application to mice may be caused by differences in the transduction efficiencies in the liver. How efficient was the deletion in the liver after AAV application?

We have quantified the deletion efficiency in the liver after the AAV application. Over 99% *Thpo* transcripts were depleted. The new data have been added to Figure 2-figure supplement 1 A in the revised manuscript.

16) In the transplantation experiment, the donor mice were treated with AAV-Cre. To address the role of hepatic Thpo in the recovery after transplantation, it would be of interest to delete Thpo in the recipients.

HSCs are defined by their capacity to reconstitute lethally irradiated recipient mice after transplantation. The purpose of the transplantation experiments is to show that there is a reduction of HSC function in mice with *Thpo* deleted from the hepatocytes by AAV treatment, independent of the surface marker profiles assayed by flow cytometry. Our data suggest that there is a reduction of HSC function in AAV-treated mice after myeloablation compared with controls.

17) The transplantation of irradiated BM cells should in general not lead to a successful engraftment of these cells. What is the aim of this experiment?

See response to question # 16. The purpose of the transplantation experiments is to confirm functional HSC reduction. We have transplanted a large number of bone marrow cells (1 million together with 1 million irradiated competitor cells) and observed significant differences between AAV- and PBS-treated mice.

18) What is the threshold of donor contribution that is judged as positive engraftment?

We set up the threshold based on background staining in recipient-type mice without transplantation in multilineage (myeloid, B and T cells, typically <0.5%). Mice with longterm multilineage reconstitution were regarded as positive engraftment.

19) As mentioned above, the study shows that, similarly to what occurs in bone marrow HSC maintenance in steady-state conditions, hepatic TPO appears to be a major functional trigger for hematopoietic recovery and HSC regeneration after myelosuppression caused by radio-chemotherapy injury.To avoid misleading conclusions and thus contribute to clinical medicine more precisely, the experimental design of the study must relate to a tumor context, since radio-chemotherapies play an established role in the treatment of cancer patients. The genetically modified animal models must have pathophysiological characteristics as comparable as possible to those that occur in humans (for animal-to-human extrapolation).We thus recommend that the reported experiments be flanked by others run upon a murine tumor model.

As the reviewer can see, our study here focuses on the identification of the cellular source of THPO for HSC regeneration and hematopoietic recovery after myeloablation. Through analyzing conditional knockout mice under two major clinically relevant myeloablative conditions (5-FU and irradiation), we showed that THPO from hepatocytes was important for HSC regeneration after myeloablation. Myeloablation is typically associated with chemo- or radio-therapy. Therefore, we believe that our findings are clinically relevant. Introducing a tumor to our experimental system may make the interpretation of the results complicated as the tumor (depending on the tumor type) may directly or indirectly contribute to the regulation of HSC regeneration. In addition, performing careful in vivo experiments on a murine tumor model using our *Thpo* conditional knockout mice as suggested by the reviewer requires extensive time and efforts. Although this is something worth investigating in the future, importantly, regardless what the results of these *Thpo* conditional knockout mice on a murine tumor model will be, they will not change our major conclusion in the current manuscript.

20) Since it has been shown that TPO is an important and nonredundant regulator of VEGF-A production in HSCs, and that is activates a VEGF-A internal autocrine loop, participating in the favorable TPO effects on HSC self-renewal and expansion (Kirito K et al., Blood. 2005; 105:4258) it would be interesting to investigate the contribution of this mechanism to the efficacy of hepatic TPO in hematopoietic recovery in stress conditions in genetically altered mouse models. This would be of interest in particular because VEGF-A, notoriously elevated in a tumor context, could also be counted among the endocrine factors which, together with TPO, contribute to bone marrow recovery, as suggested in the discussion.

Thank you for raising this interesting point regarding how THPO may promote HSC self-renewal and expansion. Kirito K et al., (Blood 2005 105:4258) show that THPO promotes VEGF-A expression in hematopoietic progenitors (Sca1+cKit+Gr1-). The authors further show that inhibition of the VEGF signaling with SU5416 (a VEGFR inhibitor) reduces the survival and proliferation of hematopoietic progenitors in response to THPO in vitro. Consistent with an earlier report (Nature 2002 417:954), the authors conclude that THPO promotes VEGF production in hematopoietic progenitors/HSCs probably through an intracellular autocrine loop. That is, the effects of VEGF are HSC-intrinsic and do not seem to depend on its secretion outside of the cell, but rather rely on its intracellular roles, since treating HSCs with cell permeable VEGFR inhibitors, but not soluble VEGFR-IgG chimeric protein, leads to their death and compromised reconstitution (Nature 2002 417:954). The focus of our current study is to elucidate the cellular source of THPO for HSC regeneration after myeloablation. The VEGF link is interesting, but it appears to be a downstream event after HSCs receive THPO signaling and probably should not be counted as an endocrine factor, particularly given that VEGF seems to act as an intracellular signal. Investigating the role of VEGF in vivo requires careful and long-term mouse genetic experiments. However, no matter what the results of these experiments may be, these studies will not impact our conclusions here. Nonetheless, we have performed qPCR to measure *Vegfa* levels in HSCs after *Thpo* was conditionally deleted from hepatocytes. We found no significant differences of *Vegfa* expression in HSCs from the conditional knockout mice compared with controls (see Author response image 2). This result is somewhat surprising. The cell purity could be a major factor as Kirito K et al., (Blood 2005 105:4258) examined Sca1+cKit+Gr1- hematopoietic progenitors while we used LSKCD150+CD48- HSCs.

**Author response image 2. sa2fig2:** Deletion of *Thpo* from hepatocytes does not lead to changes in *Vegfa* expression in HSCs. qPCR analysis showing that LSKCD150+CD48- HSCs from AAV-treated *Thpo*^*fl/fl*^ mice had similar *Vegfa* expression levels as PBS-treated controls. n = 3 mice for each condition.

21) In the discussion, in the light of their data, the authors assume that cancer patients with underlying liver disease, such as HCC, may have enhanced sensitivity to myeloablative conditioning because of liver-failure-related hepatic downregulated TPO levels. Since TPO mimetic drugs, such as romiplostim and eltrombopag, have been shown to improve recovery after ablative challenge, the authors suggest that these agents could help alleviate the adverse effects associated with myeloablative conditioning in such patients.However, it has recently been reported that, in the HCC context (Vizio B et al., Int. J. Mol. Sci. 2021, 22, 1818), the increased hepatic TPO level appears to be a novel triggering signal to promote tumor progression. Therefore, curative strategies aimed at manipulating THPOR in HCC patients should be evaluated with extreme care.The possibility of tumor conditioning should be pointed out, and recent literature in this connection included in the discussion.

Thank you for pointing out this potential issue. By no means do we suggest that promoting the THPO pathway will be universally beneficial. We have cited the paper and added discussion that strategies aimed at promoting the THPO pathway for myeloablation need careful evaluation as THPO could promote some other aspect of the pathology.